# Prognostic Impact of Heat Shock Protein 90 Expression in Women Diagnosed with Cervical Cancer

**DOI:** 10.3390/ijms25031571

**Published:** 2024-01-26

**Authors:** Tilman L. R. Vogelsang, Elisa Schmoeckel, Nicole Elisabeth Topalov, Franziska Ganster, Sven Mahner, Udo Jeschke, Aurelia Vattai

**Affiliations:** 1Department of Obstetrics and Gynecology, University Hospital, LMU Munich, 80337 Munich, Germany; 2Department of Obstetrics and Gynecology, Medical University of Graz, 8010 Graz, Austria; 3Institute of Pathology, Faculty of Medicine, LMU Munich, 80337 Munich, Germany; elisa.schmoeckel@tum.de; 4Department of Obstetrics and Gynecology, University Hospital Augsburg, 86156 Augsburg, Germany

**Keywords:** HSP90, chaperone, cervical cancer, human papillomavirus, necroptosis, programmed cell death

## Abstract

Heat Shock Protein 90 (HSP90), a major molecular chaperone, plays a crucial role in cell function by folding and stabilizing proteins and maintaining proteostasis. This study aimed to elucidate the prognostic impact of HSP90 in cervical cancer. We analyzed HSP90 expression using immunohistochemistry in cervical cancer tissue microarrays from 250 patients. This study investigated correlations between HSP90 expression levels and key clinical outcomes, including overall survival (OS), progression-free survival (PFS), and FIGO classification. The statistical analyses employed included the Kruskal–Wallis-H test, log-rank (Mantel–Cox), and Cox regression. Our findings indicate that high nuclear HSP90 expression is associated with improved OS, while high cytoplasmic HSP90 expression correlates with better PFS and a lower FIGO classification in cervical squamous cell carcinoma patients. These results suggest that HSP90 could serve as a positive prognostic factor in patients diagnosed with cervical squamous cell carcinoma, underlining its potential as a biomarker for patient prognosis and as a target for therapeutic strategies.

## 1. Introduction

Cervical cancer remains a significant global health challenge, with roughly 604,000 new cases and approximately 342,000 related deaths in 2020, ranking it as the fourth most common cancer among women worldwide [1]. This burden, however, is unevenly distributed, with a higher prevalence in developing countries due to disparities in access to screening, vaccination, and treatment. The main histological subtypes of cervical cancer are squamous cell carcinoma (accounting for about 80% of cases), adenocarcinoma (15–20%), and the less common adenosquamous carcinoma [2,3]. A key factor in the majority of cervical cancer cases is persistent infection with high-risk human papillomavirus (hr-HPV) [4], highlighting the importance of preventative measures such as HPV vaccination and effective screening programs. Upon intracellular hr-HPV replication, the viral oncoproteins E6 and E7 are expressed and promote genomic instability, hyperproliferation and immortalization, block apoptosis and disturb the cell cycle [5]. Hr-HPV also suppresses the induction of necroptosis [6]. Necroptosis, a regulated form of cell death, is critically dependent on the proteins RIPK3 (receptor-interacting serin/threonine-protein kinase 3), MLKL (mixed lineage kinase domain-like protein), and, in some cases, the kinase activity of RIPK1 (receptor-interacting serin/threonine-protein kinase 1) [7]. In the context of cervical cancer, hr-HPV can interfere with this cell death pathway. This interference occurs through the downregulation of RIPK3, leading to a partial evasion of the immune responses. This evasion potentially facilitates the progression of hr-HPV induced lesions [6]. Furthermore, research has demonstrated that the low expression of RIPK3 and RIPK1, as observed through immunohistochemical staining, is associated with a worse prognosis in cervical cancer patients. Specifically, these proteins have been identified as positive prognostic indicators for both overall survival (OS) and progression-free survival (PFS) [8].

Over the last three decades, there has been a notable decrease in the incidence and mortality rates of cervical cancer in several regions, largely attributable to the effectiveness of the Papanicolaou (Pap) smear test, the introduction of HPV vaccination, and advances in treatment programs [3,9,10]. However, it is important to recognize that this trend is primarily observed in Western countries with organized screening programs. In contrast, the majority of cervical cancer cases occur in developing countries, where access to such screening, vaccination, and advanced treatments is often limited [9,11]. The global impact of HPV vaccination, while promising, may not yet be fully realized due to its relatively recent implementation and varying coverage levels worldwide. Additionally, the introduction of HPV testing as a primary screening method has shown potential for further improving early detection, especially in countries where it has been integrated into national screening programs. This development, however, highlights the disparities in access to advanced screening techniques between developed and developing countries.

Molecular chaperones are defined as any protein that stabilizes, interacts or helps a non-native protein to gain its native conformation without being present in its final structure [12]. They are crucial for protein quality control and proteome homeostasis, also known as proteostasis, ensuring that proteins are folded correctly and functional [13]. Chaperones are upregulated in stress conditions, such as a sudden rise in temperature (hence, the term “heat shock protein”), but are also important under physiological conditions [14]. Deficiencies in proteostasis are associated with the progression and manifestation of various diseases including cancer, cardiovascular diseases, neurodegeneration, dementia and type 2 diabetes [15]. Chaperones are classified according to their molecular weight (HSP40, HSP60, HSP70, HSP90, HSP100 and small HSPs) [15].

Heat shock protein 90 (HSP90) regulates the conformation, activation, function and stability of a large number of so-called ‘client proteins’, including transcription factors and kinases [16]. It can assist protein folding, the assembly of multiprotein complexes or the binding of a ligand to its target or receptor [14]. HSP90 is also involved in several cellular processes including regulated cell death, DNA repair, development, and immune response [14,17]. During necroptosis, a complex of HSP90 and its co-chaperone cell division cycle 37 (CDC37) is required for the activation of RIPK3 [18]. Furthermore, the inhibition or disruption of HSP90 leads to the inhibition of necroptosis and the degradation of client proteins relevant to necroptosis, such as RIPK1 [19,20], RIPK3 [19,21] and MLKL [21,22].

The aim of this study was to evaluate the prognostic impact of HSP90 in cervical carcinoma patients. This focus stems from the known importance of HSP90 in the necroptosis process [17], coupled with evidence that the expression of necroptosis-related proteins is associated with improved OS and PFS in these patients and that expression of proteins relevant to necroptosis is associated with better OS and PFS [8].

## 2. Results

### 2.1. Correlation of HSP90 with Histology, Grading and TNM- and FIGO-Classification

Cytoplasmic HSP90 expression was significantly higher in adenocarcinoma and adenosqamous carcinoma (median IRS 3) compared to squamous cell carcinoma (median IRS 2) (*p* = 0.040) (as depicted in Figure 1a–d).

High cytoplasmic HSP90 expression showed a significant correlation with the pN0 status (*p* = 0.003), exhibiting a median Immuno-Reactive Score (IRS) of 2, in contrast to patients with the pN1 status, who had a median IRS of 0 (as depicted in Figure 2a). High cytoplasmic HSP90 expression was also found to significantly correlate with a low FIGO classification (*p* = 0.004), exhibiting a median IRS of 2. This contrasts with patients who have a high FIGO classification, demonstrating a median IRS of 0 (as depicted in Figure 2b).

### 2.2. High Nuclear HSP90 Expression as a Positive Prognostic Factor for OS in Patients Diagnosed with Cervical Squamous Cell Carcinoma

Patients with cervical squamous cell carcinoma exhibiting high nuclear HSP90 expression (IRS > 2) demonstrated a significantly better OS compared to those with low nuclear HSP90 expression (*p* = 0.012), as illustrated in Figure 3. Further, a multivariate Cox regression analysis, conducted on a cohort of 174 cervical squamous cell carcinoma patients, identified nuclear HSP90 expression as an independent prognostic factor for OS, as detailed in Table 1.

### 2.3. High Cytoplasmic HSP90 Expression Correlates with Improved PFS in Patients Diagnosed with Cervical Squamous Cell Carcinoma

Patients diagnosed with cervical squamous cell carcinoma and exhibiting high cytoplasmic HSP90 expression (IRS > 3) demonstrated a significantly improved PFS compared to those with low cytoplasmic HSP90 expression (*p* = 0.033), as depicted in Figure 4. Multivariate Cox regression analysis, conducted on a cohort of 168 patients with cervical squamous cell carcinoma, identified high cytoplasmic HSP90 expression as an independent positive prognostic factor for PFS, as detailed in Table 2.

### 2.4. Correlation of Nuclear and Cytoplasmic HSP90 Expression with Histopathological Parameters

Nuclear and cytoplasmic HSP90 expression were correlated with the histopathological parameters previously published in the applied cervical cancer patient collective using Spearman’s rank correlation coefficient (as shown in Appendix A). Nuclear and cytoplasmic HSP90 expression were associated strongly with each other (Spearman rho: 0.422; *p* < 0.001) and with the expression of proteins that are important for necroptosis: Nuclear HSP90 expression was significantly correlated with nuclear RIPK1 (Spearman rho: 0.309; *p* < 0.001), cytoplasmic RIPK1 (Spearman rho: 0.410; *p* < 0.001), and nuclear RIPK3 (Spearman rho: 0.517; *p* < 0.001). Correspondingly, cytoplasmic HSP90 expression was also significantly associated with these proteins, showing correlations with nuclear RIPK1 (Spearman rho: 0.226; *p* = 0.001), cytoplasmic RIPK1 (Spearman rho: 0.426; *p* < 0.001), and nuclear RIPK3 (Spearman rho: 0.422; *p* < 0.001).

Cytoplasmic HSP90 expression was found to correlate positively with cytoplasmic E6 expression (Spearman rho: 0.138; *p* = 0.042). Additionally, nuclear p53 expression was associated with both nuclear (Spearman rho: 0.208; *p* = 0.002) and cytoplasmic HSP90 expression (Spearman rho: 0.254; *p* < 0.001). Similarly, p21 expression showed a positive correlation with both nuclear (Spearman rho: 0.229; *p* = 0.002) and cytoplasmic HSP90 expression (Spearman rho: 0.180; *p* = 0.015).

Nuclear HSP90 expression was found to positively correlate with the expression of mutated p53 (Spearman rho: 0.178; *p* = 0.007), nuclear LCoR (Spearman rho: 0.181; *p* = 0.007), and glucocorticoid receptor (Spearman rho: 0.171; *p* = 0.010). Conversely, it negatively correlated with the expression of EP3 (Spearman rho: −0.192; *p* = 0.004), cytoplasmic H3K4me3 (Spearman rho: −0.215; *p* = 0.001), and MDM2 (Spearman rho: −0.156; *p* = 0.022).

Cytoplasmic HSP90 expression was positively associated with the expression of mutated p53 (Spearman rho: 0.176; *p* = 0.008), PRA in both the nucleus (Spearman rho: 0.155; *p* = 0.020) and cytoplasm (Spearman rho: 0.165; *p* = 0.013), as well as with cytoplasmic GPER (Spearman rho: 0.163; *p* = 0.016). In contrast, it was negatively associated with the expression of cytoplasmic p53 (Spearman rho: −0.150; *p* = 0.025) and mutated cytoplasmic p53 (Spearman rho: −0.205; *p* = 0.002).

## 3. Discussion

HSP90, expressed both in the nucleus and cytoplasm, plays diverse roles in cellular functions. In the nucleus, it is involved in modulating DNA structure, RNA synthesis and processing [23], and regulating transcription factors [24]. In the cytoplasm, HSP90 performs critical chaperone functions [25]. Its overexpression, which is currently under discussion for a potential role in tumorigenesis [26], has been associated with poor prognosis in cancers such as breast cancer [27,28] and colorectal cancer [29]. Consequently, HSP90 inhibitors have been explored in clinical trials for breast cancer [30,31]. Our study demonstrates that high nuclear HSP90 expression serves as an independent positive prognosticator in cervical squamous cell carcinoma. Furthermore, high cytoplasmic HSP90 expression correlates with improved PFS, lower FIGO classification, and a negative regional lymph node status (pN0). Given that HSP90 interacts with over 400 client proteins and is integral to numerous cellular pathways [32], its prognostic value could vary across different tumor entities.

In the context of cervical cancer, which is reliant on persistent hr-HPV infection [4], the viral oncoproteins E6 and E7 disrupt the cell cycle, promote immortalization, genomic instability, hyperproliferation, and inhibit apoptosis and necroptosis [5]. Previous research, including our own, has shown that the expression of necroptosis-relevant proteins (RIPK1, RIPK3) correlates with a better OS and PFS in cervical cancer patients [8]. These proteins are HSP90 clients, and inhibiting HSP90 leads to necroptosis suppression [19,33]. Our findings support this, showing a positive correlation between HSP90 expression and RIPK1 and RIPK3 in cervical cancer. While necroptosis is an undesirable mechanism in diseases like respiratory distress syndrome and chronic heart failure [34,35], in cervical cancer, it acts as a ‘backup’ mechanism, enabling infected cells to undergo regulated cell death, triggering immune responses, and potentially hindering the progression of hr-HPV-induced lesions [6]. This aligns with our key finding of a correlation between nuclear HSP90 overexpression and a better OS in cervical cancer patients.

In our study, correlations were observed between both nuclear and cytoplasmic HSP90 and nuclear p53 expression in cervical cancer, aligning with the known role of p53 as a client protein of HSP90 [36]. p53 functions as a tumor suppressor and is degraded by E6 in cervical cells with persistent hr-HPV infection [5]. Mutated p53 loses its tumor suppressor function and gains oncogenic properties [37]. The inhibition of HSP90 stabilizes mutated nuclear p53 in cancer cells [38], and HSP90 inhibitors have been shown to induce apoptosis in cervical cancer cell lines in vitro, albeit with limited efficacy in xenograft models [39]. The inhibition of HSP90 inhibits necroptosis [33]. This study’s finding that high HSP90 expression correlates with better OS in cervical cancer may seem contradictory to the growth inhibition observed with HSP90 inhibitors in vitro. This discrepancy can be reconciled by considering necroptosis as a potential ‘backup’ cell death mechanism when apoptosis is inhibited [40]. High HSP90 expression might facilitate necroptosis in hr-HPV infected cells, negating carcinogenic effects, while HSP90 inhibitors induce apoptosis with tumor-suppressive outcomes.

Moreover, we observed a negative correlation between cytoplasmic HSP90 and cytoplasmic p53, which is known for its pro-apoptotic effects [41]. This supports the notion that cytoplasmic HSP90 overexpression could inhibit apoptosis while promoting necroptosis, due to its essential chaperone role for proteins involved in necroptosis [19,20,21,22,25]. These findings suggest a dual role of HSP90 in programmed cell death: it appears to be anti-apoptotic on one hand, while potentially facilitating necroptosis on the other. Further experiments are warranted to explore these complex interactions and their implications for cervical cancer therapy.

In conclusion, our study reveals that high nuclear and cytoplasmic HSP90 expression serves as a positive prognostic indicator for OS and PFS, respectively, in patients with cervical squamous cell carcinoma. These findings underscore the dual role of HSP90 in modulating cell death pathways, particularly in the context of hr-HPV infection. The complex interplay between HSP90 and the proteins involved in apoptosis and necroptosis pathways highlights its potential as a biomarker and a therapeutic target in cervical cancer. Further research is warranted to fully understand HSP90’s multifaceted role and to explore its implications in targeted cancer therapy.

## 4. Materials and Methods

### 4.1. Characteristics of Patients and Biopsies

In this study, we included all cervical cancer patients (*n* = 250) treated surgically at the Department of Gynecology and Obstetrics, Ludwig-Maximilians-University Munich, Germany, between 1993 and 2002, for whom paraffin-embedded carcinoma tissue was available, without any pre-selection. An experienced gynecological pathologist determined the histopathological tumor subtypes (squamous cell carcinoma, adenocarcinoma, adenosquamous carcinoma) and grading (G1: well differentiated, G2: moderately differentiated, G3: poorly differentiated). TNM classification (T = primary tumor site, N = regional lymph node involvement, M = presence of distant metastatic spread) was performed in accordance with the Union of International Cancer Control (UICC) guidelines, and the FIGO classification was assessed according to the Fédération Internationale de Gynécologie et d’Obstétrique (FIGO) criteria. Clinical and patient follow-up data were obtained from the Munich Cancer Registry (as presented in Table 3). The mean age of the patients at diagnosis was 49.0 ± 13.0 years, ranging from 20.4 to 83.3 years. Patients with an unknown cause of death or death unrelated to cervical cancer were censored. Correlation analyses with HSP90 expression were enabled by evaluating the other histopathological parameters in various previously published papers.

### 4.2. Immunohistochemistry

The cervical cancer tissue samples were fixed in 3.7% neutral buffered formalin immediately after resection, and then embedded in paraffin using standard procedures. Tissue microarrays (TMAs) were created from these formalin-fixed, paraffin-embedded tissues. For the immunohistochemical analysis, 3 μm tissue sections were first deparaffinized in Roticlear (Carl Roth GmbH + Co. KG, Karlsruhe, Germany) for 20 min, briefly washed in 100% ethanol, and then immersed in 3% H_2_O_2_ in methanol at room temperature for 20 min to inactivate the endogenous peroxidase. After rehydration through a descending ethanol gradient and a brief rinse in distilled water, the sections underwent epitope retrieval in a pressure cooker for 5 min using a sodium citrate buffer (pH 6.0; 0.1 mol/L citric acid/0.1 mol/L sodium citrate). Post retrieval, the slides were washed in distilled water and phosphate-buffered saline (PBS) twice for 2 min each. To block non-specific binding, Blocking Solution (ZytoChem Plus HRP Polymer System (Mouse/Rabbit); Zytomed Systems GmbH, Berlin, Germany) was applied for 5 min at room temperature. The sections were then incubated overnight at 4 °C with anti-HSP90 antibody (monoclonal mouse IgG, ab13492; Abcam, Cambridge, UK) diluted 1:1000 in PBS. This was followed by incubation with PostBlock Reagent and HRP-Polymer (Mouse/Rabbit) for 20 and 30 min, respectively, which contained secondary antibodies and peroxidase. After each incubation step, the slides were washed with PBS. For visualization, 3,3′-diaminobenzidine chromogen (DAB; Dako, Glostrup, Denmark) and its substrate buffer were applied for 30 s, and the reaction was halted by immersing the slides in distilled water. The tissue was counterstained with Mayer’s acidic hematoxylin, dehydrated in an ascending ethanol gradient and Roticlear, and finally cover slipped with ROTI^®^ Mount (Carl Roth GmbH + Co. KG, Karlsruhe, Germany). Fallopian tube tissue served as a positive control. Negative controls were processed by replacing primary antibodies with specific isotype control antibodies (BioGenex, Fremont, CA, USA).

### 4.3. Quantification

For the examination of the immunohistochemically stained cervical cancer slides, a Leitz Diaplan photomicroscope (Leitz, Wetzlar, Germany) was used. The staining of each specimen was quantified by the application of the semi-quantitative immunoreactive score (IRS), which is used for the optical evaluation of the intensity and distribution pattern of antigen expression [42]. The IRS was calculated by multiplying the number of positively stained cells (in %) (0: no staining; 1: 1–10% stained tumor cells; 2: 11–50% stained tumor cells; 3: 51–80% stained tumor cells; 4: >80% stained tumor cells) with the predominant staining intensity (0: none; 1: weak; 2: moderate; 3: strong). The scale goes from 0 (no expression) to 12 (very high expression). Images were taken with Flexacam C1 (Leica Microsystems (Switzerland) Ltd., Heerbrugg, Switzerland).

### 4.4. Statistical Analysis

IBM SPSS Statistics Version 26.0.0.0 (IBM, Armonk, New York, NY, USA) was used for data analysis. *p*-values of *p* < 0.05 were considered statistically significant. Group comparisons of independent groups regarding the clinical and pathological subgroups were tested with the Kruskal–Wallis–H test. Spearman’s rank correlation coefficient was applied for the bivariate correlations between staining results in this study and the histopathological variables. The OS and PFS of the cervical cancer patients were tested for significance using the log-rank (Mantel-Cox) test and compared using Kaplan–Meier curves. A Cox regression model was established for a multivariate analysis of the investigated parameters. The variables included in the Cox regression model were the patient’s age, histological subtype, tumor grading, TNM and FIGO classification, and HSP90 expression. HSP90 expression was divided into low and high expression for survival analysis. During analyses, the observers were fully blinded to the patients’ data.

## Figures and Tables

**Figure 1 ijms-25-01571-f001:**
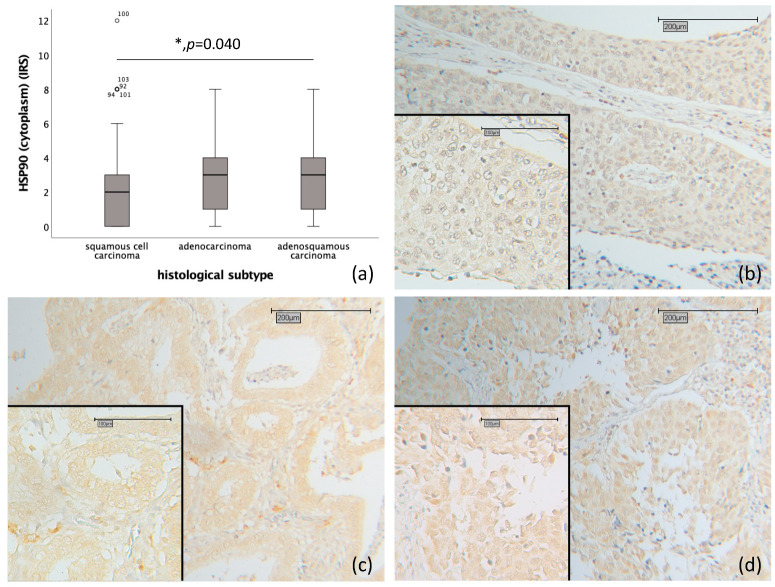
Correlation of cytoplasmic HSP90 expression with histological subtype in cervical carcinoma (*p* = 0.040). (**a**) Boxplot of cytoplasmic HSP90 expression and histological subtype in cervical carcinoma. (**b**) Cervical carcinoma and squamous cell carcinoma (*n* = 180) with a cytoplasmic HSP90 IRS of 1; magnification ×10 and ×25 in the inset. (**c**) Cervical carcinoma and adenocarcinoma (*n* = 32) with a cytoplasmic HSP90 IRS of 4; magnification ×10 and ×25 in the inset. (**d**) Cervical carcinoma and adenosquamous carcinoma (*n* = 13) with a cytoplasmic HSP90 IRS of 4; magnification ×10 and ×25 in the inset.

**Figure 2 ijms-25-01571-f002:**
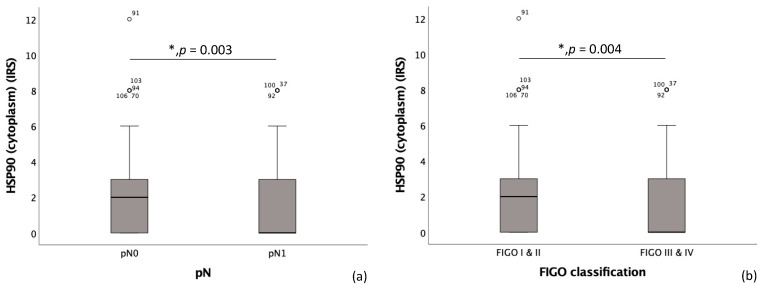
Correlation of HSP90 with pN and FIGO-classification. (**a**) Boxplot of cytoplasmic HSP90 expression and regional lymph node status (pN) in cervical carcinoma (*p* = 0.003); median IRS of cytoplasmic HSP90 expression in pN0 is 2 (*n* = 125) and in pN1 is 0 (*n* = 94). (**b**) Boxplot of cytoplasmic HSP90 expression and FIGO classification in cervical carcinoma (*p* = 0.004); median IRS of cytoplasmic HSP90 expression in FIGO I and II is 2 (*n* = 125) and in FIGO III and IV is 0 (*n* = 125).

**Figure 3 ijms-25-01571-f003:**
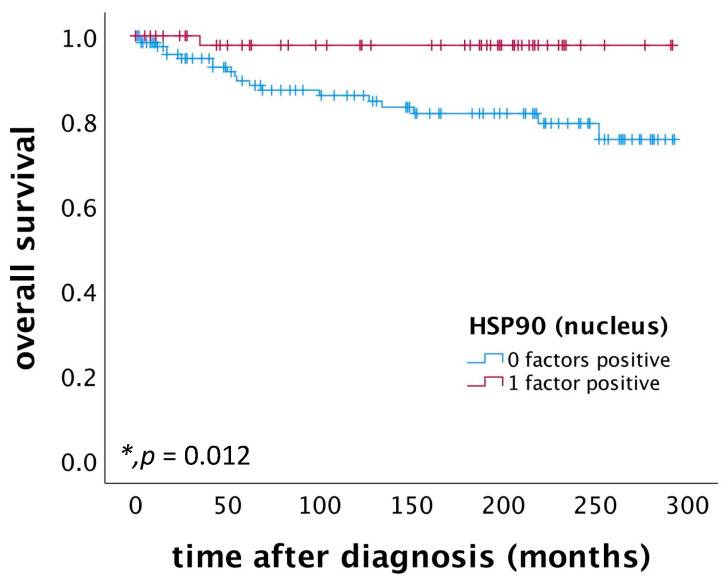
OS of patients diagnosed with cervical squamous cell carcinoma correlated with nuclear HSP90. Nuclear HSP90 expression (*n* = 53) is associated with a better OS in cervical squamous cell carcinoma patients compared to patients not expressing nuclear HSP90 (*n* = 127) (*p* = 0.012).

**Figure 4 ijms-25-01571-f004:**
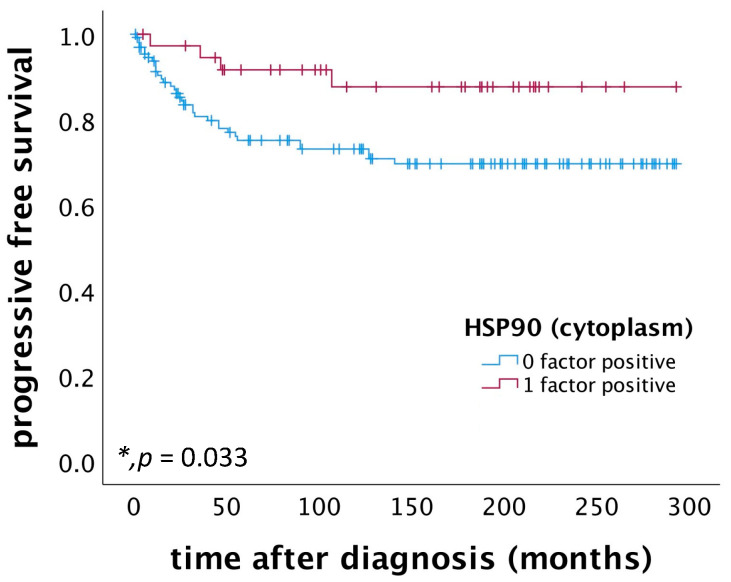
PFS in patients diagnosed with cervical squamous cell carcinoma was found to be associated with cytoplasmic HSP90 expression. Specifically, patients exhibiting high cytoplasmic HSP90 expression (*n* = 38) demonstrated significantly improved PFS compared to those with low cytoplasmic HSP90 expression (*n* = 133) (*p* = 0.033).

**Table 1 ijms-25-01571-t001:** Multivariate Cox regression analysis of cervical squamous cell carcinoma patients (*n* = 174) and their clinical and pathological characteristics including the nuclear HSP90 expression regarding OS. Significant independent factors for OS are indicated with asterisks. (* *p* < 0.05).

Covariate	Hazard Ratio	95% CI	*p*-Value
patient’s age (<49 years vs. ≥49 years)	1.372	0.540–3.488	0.507
tumor grading	0.986	0.895–1.086	0.769
extent of the primary tumor [7]	2.510	1.048–6.015	0.039 *
nodal status (pNX/0 vs. pN1)	2.311	0.813–6.570	0.116
FIGO classification	0.407	0.171–0.971	0.043 *
positive nuclear HSP90 expression	0.127	0.017–0.957	0.045 *

**Table 2 ijms-25-01571-t002:** Multivariate Cox regression analysis of cervical squamous cell carcinoma patients (*n* = 168) and their clinical and pathological characteristics including cytoplasmic HSP90 expression regarding PFS. Significant independent factors for PFS are indicated with asterisks. (* *p* < 0.05).

Covariate	Hazard Ratio	95% CI	*p*-Value
patient’s age (<49 years vs. ≥49 years)	1.032	0.525–2.030	0.927
tumor grading	0.986	0.902–1.077	0.751
extent of the primary tumor [7]	2.104	1.096–4.038	0.025 *
nodal status (pNX/0 vs. pN1)	2.438	1.167–5.091	0.018 *
FIGO classification	0.496	0.259–0.950	0.034 *
positive cytoplasmic HSP90 expression	0.316	0.108–0.923	0.035 *

**Table 3 ijms-25-01571-t003:** Description of the clinical and pathological variables of the patients.

	Number of Cases (Total Number of Cases: *n* = 250)	%
Histopathological tumor subtype		
squamous cell carcinoma	194	77.6
adenocarcinoma	34	13.6
adenosquamous carcinoma	15	6.0
unknown	7	2.8
Tumor grading		
G1	21	8.4
G2	143	57.2
G3	78	31.2
unknown	8	3.2
Extent of primary tumor [7]		
pT1	107	42.8
pT2	126	50.4
pT3	9	3.6
pT4	1	0.4
unknown	7	2.8
Regional lymph node involvement (pN)		
pN0	145	58.0
pN1	98	39.2
unknown	7	2.8
Presence of distant metastatic spread (pM)		
pM0	2	0.8
pM1	7	2.8
pMX	235	94
unknown	6	2.4
FIGO classification		
FIGO I	79	31.6
FIGO II	64	25.6
FIGO III	93	37.2
FIGO IV	7	2.8
unknown	7	2.8
Progression		
none	180	72.0
at least one	63	25.2
unknown	7	2.8
Survival		
censured	211	84.4
dead (tumor-dependent)	33	13.2
unknown	6	2.4

## Data Availability

Data are contained within the article.

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
