# Peer review of "Prognostic Impact of Heat Shock Protein 90 Expression in Women Diagnosed with Cervical Cancer"

_ijms, 2024, doi:10.3390/ijms25031571_

Round 1
Reviewer 1 Report
Comments and Suggestions for Authors
In my humble opinion, the manuscript is well written, the idea of ​​the research is clear, it is understandable; and it has originality, significance, in general it is well presented and is of interest to readers.
Well, they highlight that if there is no Hsp90 or it is inhibited, necroptosis is inhibited and the degradation of proteins, such as RIPK1, etc.; What they see improves the survival prognosis.
Also in the results it is clear, at least in the test of independent predictors for Overal Survival, that there are values ​​like those of at least the other two independent factors (extent of the primary tumor; and FIGO classification). Likewise, they also show convincing data about the relationship between elevated levels of cytoplasmic Hsp90 and Progressive Free Survival. They also demonstrate that nuclear and cytoplasmic expressions of Hsp90 are strongly associated with each other; and with the expression of proteins important for Necroptosis, such as RIPK1; and its nuclear and cytoplasmic expression. And finally the cytoplasmic expression of Hsp90 as it correlates with the expression of the E6 oncoprotein. In general, their data support the idea that overexpression of cytoplasmic Hsp90 could inhibit apoptosis and simultaneously facilitate necroptosis.
I would recommend accepting the publication of this manuscript, however, and based on the other comments from the reviewers, I would suggest correcting the cervcical word, which is misspelled on page 4; in the last line of point 2.2. Just as please excuse the observation, but it is clear to me in figure 3 why they write down the legend *,p=0.012; of which I also do not see that they refer to it anywhere in the text.
Reviewer 2 Report
Comments and Suggestions for Authors
Vogelsang et al. have explored the prognostic impact of Heat Shock Protein 90 (HSP90) in cervical squamous cell carcinoma. This research involved the examination of HSP90 expression in tissue samples from 250 cervical cancer patients, employing immunohistochemistry techniques. The study's goal was to correlate the levels of HSP90 expression with various outcomes such as overall survival, progression-free survival, and clinical outcomes. Advanced statistical methods like the Kruskal-Wallis-H test, log-rank (Mantel-Cox), and Cox regression were utilized for data analysis.
The results highlighted a significant correlation between high nuclear HSP90 expression and enhanced overall survival rates in patients with cervical squamous cell carcinoma. Similarly, increased cytoplasmic HSP90 expression was associated with better progression-free survival. Furthermore, a lower FIGO classification, indicative of less severe disease, was linked to higher cytoplasmic HSP90 expression. These findings suggest that HSP90 plays a positive and impactful role in the prognosis of cervical squamous cell carcinoma.
The claims are properly placed in the context of the previous literature. The experimental data support the claims. The manuscript is written clearly enough that most of it is understandable to non-specialists. The authors have provided adequate proof for their claims, without overselling them. The authors have treated the previous literature fairly. The paper offers enough details of methodology so that the experiments could be reproduced.
Comments
Vogelsang et al. write in the introduction:
Line 42-44, "Over the last three decades, worldwide incidence and death of cervical cancer has slowly decreased due to the effectiveness of the Papanicolaou (Pap) - smear test and the introduction of vaccination and effective treatment programs"
This is not compleatly true. While the introduction of the Papanicolaou (Pap) smear test has indeed led to a reduction in the incidence and mortality of cervical cancer, this benefit is predominantly seen in Western countries with organized screening programs. It's important to note that the majority of cervical cancer cases occur in developing countries, where such screening programs are often limited or non-existent. This discrepancy results in a skewed representation of the global situation, where reductions in incidence and mortality are not uniformly observed across all regions.
The world-wide coverage of Human Papillomavirus (HPV) vaccination is still limited. Furthermore, given the time lag – typically 10-15 years – between HPV vaccination and its potential impact on reducing cervical cancer rates, it might be premature to attribute a significant global reduction in cervical cancer incidence to the HPV vaccine. This is particularly relevant as the vaccine has not been in use long enough on a global scale to manifest a marked decrease in cervical cancer rates.
In many developing countries, access to effective medical treatment for cervical cancer is limited. This lack of access significantly influences the overall global statistics for cervical cancer incidence and mortality. In these regions, even if screening programs identify cervical cancer, the subsequent steps in management and treatment are often not as accessible or effective as they are in more developed countries.
Another significant advancement in the prevention of cervical cancer, alongside HPV vaccination, is the introduction of Human Papillomavirus (HPV) testing as a primary screening method. Some Western countries have already integrated HPV testing into their cervical cancer screening programs. This method is more sensitive than the traditional Pap smear test and can identify women at higher risk of cervical cancer earlier and more accurately.
Despite the proven effectiveness of HPV testing, there is a stark contrast in its availability and implementation between developed and developing countries. Most developing countries lack access to such high-performance screening tests. This disparity not only affects the early detection and subsequent management of cervical cancer but also widens the gap in cervical cancer outcomes between developed and developing nations.
I suggest this formulation:
"Over the last three decades, there has been a notable decrease in the incidence and mortality rates of cervical cancer in several regions, largely attributable to the effectiveness of the Papanicolaou (Pap) smear test, the introduction of HPV vaccination, and advances in treatment programs. However, it is important to recognize that this trend is primarily observed in Western countries with organized screening programs. In contrast, the majority of cervical cancer cases occur in developing countries, where access to such screening, vaccination, and advanced treatments is often limited. The global impact of HPV vaccination, while promising, may not yet be fully realized due to its relatively recent implementation and varying coverage levels worldwide. Additionally, the introduction of HPV testing as a primary screening method has shown potential for further improving early detection, especially in countries where it has been integrated into national screening programs. This development, however, highlights the disparities in access to advanced screening techniques between developed and developing countries."
Minor revisions
Line 2-3, title, "Prognostic Impact of Heat Shock Protein 90 Expression in Women Diagnosed with Cervical Cancer"
Line 11-23, abstract, "Abstract: Heat Shock Protein 90 (HSP90), a major molecular chaperone, plays a crucial role in cell function by folding and stabilizing proteins and maintaining proteostasis. This study aimed to elucidate the prognostic impact of HSP90 in cervical cancer. We analyzed HSP90 expression using immunohistochemistry in cervical cancer tissue microarrays from 250 patients. The study investigated correlations between HSP90 expression levels and key clinical outcomes, including overall survival (OS), progression-free survival (PFS), and FIGO classification. Statistical analyses employed included the Kruskal-Wallis-H test, log-rank (Mantel-Cox), and Cox regression. Our findings indicate that high nuclear HSP90 expression is associated with improved OS, whereas high cytoplasmic HSP90 expression correlates with better PFS and lower FIGO classification in cervical squamous cell carcinoma patients. These results suggest that HSP90 could serve as a positive prognostic factor in patients diagnosed with cervical squamous cell carcinoma, underlining its potential as a biomarker for patient prognosis and as a target for therapeutic strategies."
Line 28-33, introduction, "Cervical cancer remains a significant global health challenge, with roughly 604,000 new cases and approximately 342,000 related deaths in 2020, ranking it as the fourth most common cancer among women worldwide [1]. This burden, however, is unevenly distributed, with a higher prevalence in developing countries due to disparities in access to screening, vaccination, and treatment. The main histological subtypes of cervical cancer are squamous cell carcinoma (accounting for about 80% of cases), adenocarcinoma (15-20%), and the less common adenosquamous carcinoma [2, 3]. A key factor in the majority of cervical cancer cases is persistent infection with high-risk human papillomavirus (hr-HPV) [4], highlighting the importance of preventative measures such as HPV vaccination and effective screening programs."
Line 36-42, introduction, "Necroptosis, a regulated form of cell death, is critically dependent on the proteins RIPK3, MLKL, and, in some cases, the kinase activity of RIPK1 [7]. In the context of cervical cancer, high-risk human papillomavirus (hr-HPV) can interfere with this cell death pathway. This interference occurs through the downregulation of RIPK3, leading to a partial evasion of the immune responses. This evasion potentially facilitates the progression of hr-HPV induced lesions [6]. Furthermore, research has demonstrated that low expression of RIPK3 and RIPK1, as observed through immunohistochemical staining, is associated with a worse prognosis in cervical cancer patients. Specifically, these proteins have been identified as negative prognostic indicators for both overall survival (OS) and progression-free survival (PFS) [8]."
Line 64-66, introduction, "The aim of this study was to evaluate the prognostic impact of HSP90 in cervical carcinoma patients. This focus stems from the known importance of HSP90 in the necroptosis process [16], coupled with evidence that the expression of necroptosis-related proteins is associated with improved overall survival (OS) and progression-free survival (PFS) in these patients [8]."
Line 73-74, results, "High cytoplasmic HSP90 expression showed a significant correlation with pN0 status (p = 0.003), exhibiting a median Immuno-Reactive Score (IRS) of 2, in contrast to patients with pN1 status, who had a median IRS of 0 (as depicted in Figure 2a)."
Line 75-77, results, "High cytoplasmic HSP90 expression was also found to significantly correlate with a low FIGO classification (p = 0.004), exhibiting a median Immuno-Reactive Score (IRS) of 2. This contrasts with patients who have a high FIGO classification, demonstrating a median IRS of 0 (as depicted in Figure 2b)."
Line 94-102, results, "2.2. High nuclear HSP90 expression as a positive prognostic factor for OS in patients diagnosed with cervical squamous cell carcinoma
Patients with cervical squamous cell carcinoma exhibiting high nuclear HSP90 expression (IRS > 2) demonstrated significantly better overall survival (OS) compared to those with low nuclear HSP90 expression (p = 0.012), as illustrated in Figure 3.
Further, multivariate Cox regression analysis, conducted on a cohort of 174 cervical squamous cell carcinoma patients, identified nuclear HSP90 expression as an independent prognostic factor for OS, as detailed in Table 1."
Line 111-115, results, "2.3. High cytoplasmic HSP90 expression correlates with improved PFS in patients diagnosed with cervical squamous cell carcinoma
Patients diagnosed with cervical squamous cell carcinoma and exhibiting high cytoplasmic HSP90 expression (IRS > 3) demonstrated significantly improved progression-free survival (PFS) compared to those with low cytoplasmic HSP90 expression (p = 0.033), as depicted in Figure 4."
Line 116-119, results, "Multivariate Cox regression analysis, conducted on a cohort of 168 patients with cervical squamous cell carcinoma, identified high cytoplasmic HSP90 expression as an independent positive prognostic factor for progression-free survival (PFS), as detailed in Table 2."
Line 122-124, Figure 4, "Progression-free survival (PFS) in patients diagnosed with cervical squamous cell carcinoma was found to be associated with cytoplasmic HSP90 expression. Specifically, patients exhibiting high cytoplasmic HSP90 expression (n = 38) demonstrated significantly improved PFS compared to those with low cytoplasmic HSP90 expression (n = 133) (p = 0.033)."
Line 136-141, results, "Nuclear HSP90 expression was significantly correlated with nuclear RIPK1 (Spearman rho: 0.309; p < 0.001), cytoplasmic RIPK1 (Spearman rho: 0.410; p < 0.001), and nuclear RIPK3 (Spearman rho: 0.517; p < 0.001). Correspondingly, cytoplasmic HSP90 expression was also significantly associated with these proteins, showing correlations with nuclear RIPK1 (Spearman rho: 0.226; p = 0.001), cytoplasmic RIPK1 (Spearman rho: 0.426; p < 0.001), and nuclear RIPK3 (Spearman rho: 0.422; p < 0.001)."
Line 142-147, results, "Cytoplasmic HSP90 expression was found to correlate positively with cytoplasmic E6 expression (Spearman rho: 0.138; p < 0.042). Additionally, nuclear p53 expression was associated with both nuclear (Spearman rho: 0.208; p = 0.002) and cytoplasmic HSP90 expression (Spearman rho: 0.254; p < 0.001). Similarly, p21 expression showed a positive correlation with both nuclear (Spearman rho: 0.229; p = 0.002) and cytoplasmic HSP90 expression (Spearman rho: 0.180; p = 0.015)."
Line 148-152, results, "Nuclear HSP90 expression was found to positively correlate with the expression of mutated p53 (Spearman rho: 0.178; p = 0.007), nuclear LCoR (Spearman rho: 0.181; p = 0.007), and glucocorticoid receptor (Spearman rho: 0.171; p = 0.010). Conversely, it negatively correlated with the expression of EP3 (Spearman rho: -0.192; p = 0.004), cytoplasmic H3K4me3 (Spearman rho: -0.215; p = 0.001), and MDM2 (Spearman rho: -0.156; p = 0.022)."
Line 153-157, results, "Cytoplasmic HSP90 expression was positively associated with the expression of mutated p53 (Spearman rho: 0.176; p = 0.008), PRA in both the nucleus (Spearman rho: 0.155; p = 0.020) and cytoplasm (Spearman rho: 0.165; p = 0.013), as well as with cytoplasmic GPER (Spearman rho: 0.163; p = 0.016). In contrast, it was negatively associated with the expression of cytoplasmic p53 (Spearman rho: -0.150; p = 0.025) and mutated cytoplasmic p53 (Spearman rho: -0.205; p = 0.002)."
Line 160-186, discussion, "HSP90, expressed both in the nucleus and cytoplasm, plays diverse roles in cellular functions. In the nucleus, it is involved in modulating DNA structure, RNA synthesis and processing [22], and regulating transcription factors [23]. In the cytoplasm, HSP90 performs critical chaperone functions [24]. Its overexpression, which is currently under discussion for a potential role in tumorigenesis [25], has been associated with poor prognosis in cancers such as breast [26, 27] and colorectal cancer [28]. Consequently, HSP90 inhibitors have been explored in clinical trials for breast cancer [29, 30]. Our study demonstrates that high nuclear HSP90 expression serves as an independent positive prognosticator in cervical squamous cell carcinoma, contrasting patients with low nuclear HSP90 expression. Furthermore, high cytoplasmic HSP90 expression correlates with improved progression-free survival (PFS), lower FIGO classification, and negative regional lymph node status (pN0). Given that HSP90 interacts with over 400 client proteins and is integral to numerous cellular pathways [31], its prognostic value could vary across different tumor types.
In the context of cervical cancer, which is reliant on persistent hr-HPV infection [4], the viral oncoproteins E6 and E7 disrupt the cell cycle, promote immortalization, genomic instability, hyperproliferation, and inhibit apoptosis and necroptosis [5]. Previous research, including our own, has shown that expression of necroptosis-relevant proteins (RIPK1, RIPK3) correlates with better OS and PFS in cervical cancer patients [8]. These proteins are HSP90 clients, and inhibiting HSP90 leads to necroptosis suppression [18, 32]. Our findings support this, showing a positive correlation between HSP90 expression and RIPK1 and RIPK3 in cervical cancer. While necroptosis is an undesirable mechanism in diseases like respiratory distress syndrome and chronic heart failure [33, 34], in cervical cancer, it acts as a 'backup' mechanism, enabling infected cells to undergo regulated cell death, triggering immune responses, and potentially hindering the progression of hr-HPV-induced lesions [6]. This aligns with our key finding of a correlation between nuclear HSP90 overexpression and better OS in cervical cancer patients."
Line 187-213, discussion, "In our study, correlations were observed between both nuclear and cytoplasmic HSP90 and nuclear p53 expression in cervical cancer, aligning with the known role of p53 as a client protein of HSP90 [35]. p53 functions as a tumor suppressor and is degraded by E6 in cervical cells with persistent hr-HPV infection [5]. Mutated
p53 loses its tumor suppressor function and gains oncogenic properties [36]. Inhibition of HSP90 stabilizes mutated nuclear p53 in cancer cells [37], and HSP90 inhibitors have been shown to induce apoptosis in cervical cancer cell lines in vitro, albeit with limited efficacy in xenograft models [38].
This study's finding that high HSP90 expression correlates with better overall survival (OS) in cervical cancer may seem contradictory to the growth inhibition observed with HSP90 inhibitors in vitro. This discrepancy can be reconciled by considering necroptosis as a potential 'backup' cell death mechanism when apoptosis is inhibited [39]. High HSP90 expression might facilitate necroptosis in hr-HPV infected cells, negating carcinogenic effects, while HSP90 inhibitors induce apoptosis with tumor-suppressive outcomes.
Moreover, we observed a negative correlation between cytoplasmic HSP90 and cytoplasmic p53, which is known for its pro-apoptotic effects [40]. This supports the notion that cytoplasmic HSP90 overexpression could inhibit apoptosis while promoting necroptosis, due to its essential chaperone role for proteins involved in necroptosis [18-21, 24]. These findings suggest a dual role of HSP90 in programmed cell death: it appears to be anti-apoptotic on one hand, while potentially facilitating necroptosis on the other. Further experiments are warranted to explore these complex interactions and their implications for cervical cancer therapy."
Line 213, add a conclusion, "In conclusion, our study reveals that high nuclear and cytoplasmic HSP90 expression serves as a positive prognostic indicator for overall survival and progression-free survival, respectively, in patients with cervical squamous cell carcinoma. These findings underscore the dual role of HSP90 in modulating cell death pathways, particularly in the context of hr-HPV infection. The complex interplay between HSP90 and proteins involved in apoptosis and necroptosis pathways highlights its potential as a biomarker and a therapeutic target in cervical cancer. Further research is warranted to fully understand HSP90's multifaceted role and to explore its implications in targeted cancer therapy."
Line 216-231, materials and methods, "In this study, we included all cervical cancer patients (n = 250) treated surgically at the Department of Gynecology and Obstetrics, Ludwig-Maximilians-University Munich, Germany, between 1993 and 2002, for whom paraffin-embedded carcinoma tissue was available, without any pre-selection. An experienced gynecological pathologist determined histopathological tumor subtypes (squamous cell carcinoma, adenocarcinoma, adenosquamous carcinoma) and grading (G1: well differentiated, G2: moderately differentiated, G3: poorly differentiated). TNM classification (T = primary tumor site, N = regional lymph node involvement, M = presence of distant metastatic spread) was performed in accordance with the Union of International Cancer Control (UICC) guidelines, and FIGO classification was assessed according to the Fédération Internationale de Gynécologie et d'Obstétrique (FIGO) criteria. Clinical and patient follow-up data were obtained from the Munich Cancer Registry (as presented in Table 2). The mean age of the patients at diagnosis was 49.0 ± 13.0 years, ranging from 20.4 to 83.3 years. Patients with an unknown cause of death or death unrelated to cervical cancer were censored. Correlation analyses with HSP90 expression were enabled by evaluating other histopathological parameters in various previously published papers."
Line 235-261, immunohistochemistry, "Cervical cancer tissue samples were fixed in 3.7% neutral buffered formalin immediately after resection, and then embedded in paraffin using standard procedures. Tissue microarrays (TMAs) were created from these formalin-fixed, paraffin-embedded tissues. For immunohistochemical analysis, 3μm tissue sections were first deparaffinized in Roticlear (Carl Roth GmbH + Co. KG, Karlsruhe, Germany) for 20 minutes, briefly washed in 100% ethanol, and then immersed in 3% H2O2 in methanol at room temperature for 20 minutes to inactivate endogenous peroxidase. After rehydration through a descending ethanol gradient and a brief rinse in distilled water, the sections underwent epitope retrieval in a pressure cooker for 5 minutes using a sodium citrate buffer (pH 6.0; 0.1 mol/L citric acid/ 0.1 mol/L sodium citrate).
Post-retrieval, the slides were washed in distilled water and phosphate-buffered saline (PBS) twice for 2 minutes each. To block non-specific binding, Blocking Solution (ZytoChem Plus HRP Polymer System (Mouse/Rabbit); Zytomed Systems GmbH, Berlin, Germany) was applied for 5 minutes at room temperature. The sections were then incubated overnight at 4°C with anti-HSP90 antibody (monoclonal mouse IgG, ab13492; Abcam, Cambridge, UK) diluted 1:1000 in PBS. This was followed by incubation with PostBlock Reagent and HRP-Polymer (Mouse/Rabbit) for 20 and 30 minutes, respectively, which contained secondary antibodies and peroxidase. After each incubation step, the slides were washed with PBS.
For visualization, 3,3′-diaminobenzidine chromogen (DAB; Dako, Glostrup, Denmark) and its substrate buffer were applied for 30 seconds, and the reaction was halted by immersing the slides in distilled water. The tissue was counterstained with Mayer’s acidic hematoxylin, dehydrated in an ascending ethanol gradient and Roticlear, and finally cover slipped with ROTI® Mount (Carl Roth GmbH + Co. KG, Karlsruhe, Germany). Fallopian tube tissue served as a positive control. Negative controls were processed by replacing primary antibodies with specific isotype control antibodies (BioGenex, Fremont, CA, USA)."
Comments on the Quality of English Language
As a non-native English speaker, I have suggested some reformulations to improve the language in the manuscript. However, I recommend considering a review by a native English speaker to further enhance sentence construction, readability, coherence, and accuracy, ensuring a polished final presentation.
